# A Critical Review of Deep Learning-Based Multi-Sensor Fusion Techniques

**DOI:** 10.3390/s22239364

**Published:** 2022-12-01

**Authors:** Benedict Marsh, Abdul Hamid Sadka, Hamid Bahai

**Affiliations:** 1Institute of Digital Futures, Brunel University London, Kingston Ln, Uxbridge UB8 3PH, UK; 2Institute of Materials and Manufacturing, Brunel University London, Kingston Ln, Uxbridge UB8 3PH, UK

**Keywords:** sensor fusion, stereo, LiDAR, deep learning

## Abstract

In this review, we provide a detailed coverage of multi-sensor fusion techniques that use RGB stereo images and a sparse LiDAR-projected depth map as input data to output a dense depth map prediction. We cover state-of-the-art fusion techniques which, in recent years, have been deep learning-based methods that are end-to-end trainable. We then conduct a comparative evaluation of the state-of-the-art techniques and provide a detailed analysis of their strengths and limitations as well as the applications they are best suited for.

## 1. Introduction

Multi-sensor fusion is the process of combining data that have been captured from multiple sensors. The data from the different sensors will be fused into a single representation of the scene that is more accurate than if it were to be computed using any one of the input data alone.

These fusion techniques can be applied to generate an accurate 3D representation of a real-world scene using multiple sensors. The fusion of the sensor data will allow more accurate 3D models in environments where one sensor is less accurate. In this paper, the techniques that we will consider use RGB cameras and a LiDAR sensor and combine the data in a single dense depth map.

The RGB cameras will capture two stereo images and the LiDAR sensor will capture a 3D point cloud. The point cloud will be projected onto one of the RGB images and the fusion method will be used to combine these data into a single dense depth map.

The depth computed from the stereo will be less reliable in environments with high occlusion and a lack of textured surfaces. LiDAR sensors will capture point clouds which will result in sparse depth maps when projected. These modalities can be combined into a depth map which is dense and can be more accurate in an environment with a lack of occlusion and less texture, which is useful for autonomous driving applications.

The depth map can be computed using the two RGB cameras with stereo matching to obtain a disparity map which can be converted into a depth map with the camera parameters. Recently, the most predominant techniques for computing disparity from rectified stereo images have been deep learning-based methods [1,2,3,4,5,6,7,8,9,10,11], starting with the MC-CNN [12], and then extended-use end-to-end trainable networks, starting with DispNet [13].

The depth map can also be computed by using the LiDAR-projected sparse depth map and using its corresponding RGB image as guidance to compute the dense depth map. The recent methods for depth completion from sparse depth maps are mostly deep learning-based methods [14,15,16,17,18,19,20,21,22]. These methods usually use extra input data modalities as well as the sparse LiDAR, such as an RGB image or a semantic map.

The depth map can be computed using both the RBG stereo images and the sparse LiDAR depth map. The recent methods are deep learning based [23,24,25] and use similar network architectures to the deep learning-based stereo methods.

The other methods we will not consider are computing the depth map from a single RGB image and using only the sparse LiDAR depth map because we are only covering the methods that use data from multiple sensors for the data fusion.

## 2. State-of-the-Art Multi-Sensor Fusion Techniques and Their Performance Evaluation

In this section, we review the state-of-the-art methods used for computing the disparity or depth maps using RGB and LiDAR. We evaluate the performance of the method using the metrics reported and the visual results from the depth or disparity maps computed, using colour to visualise the depth or disparity values. The methods can be categorised by the input data they use to compute the fused depth map as shown in Table 1. The main types are using stereo images as the input data, using a LiDAR-projected depth map and stereo images or using a LiDAR-projected depth map and a single RGB image.

The methods that use stereo or LiDAR and stereo as the input data follow a similar technique. They use a feature network to extract the feature maps from both of the stereo images. Then, by matching the features from one of the stereo images with the features from the other stereo image, a matching cost can be computed for each possible disparity. Then, further networks can be used to compute the final disparity map prediction, and if it is a LiDAR and stereo method, the networks will use the LiDAR data as an extra input to improve the accuracy when computing the matching cost.

The methods that use LiDAR and a single RGB image as input data also follow a similar technique of using a network to compute an initial coarse depth map prediction. This prediction is then refined using features from the network to compute multiple updates to the depth map which will increase the accuracy of the depth map after each update.

### 2.1. Raft-Stereo: Multilevel Recurrent Field Transforms for Stereo Matching

This method [26] computes the disparity from only RGB stereo images using an approach similar to RAFT [35]. First, the feature maps are extracted from the left and right images using a feature network which is a convolutional neural network. Then, the feature maps are combined into a correlation volume by taking the dot product of each possible feature pair corresponding to a possible disparity. A correlation pyramid is constructed by repeatedly downsampling the dimension corresponding to the disparity in the correlation volume to obtain multiple correlation volumes.

An initial prediction of 0 disparity for all pixels is used as a starting point on the disparity map. The initial disparity map is updated by using a recurrent neural network. To obtain the inputs for the recurrent network, the correlation pyramid is indexed by selecting windows in each volume around the current disparity estimates. The windows are concatenated together into a single feature map which is then fed through convolutional layers to obtain the correlation features, and the current disparity prediction is also fed through convolutional layers to obtain the disparity features. The left image is fed through a context network to extract the context features. The correlation, disparity and context features are concatenated into a single feature map and used as the input for a multi-scale GRU.

The multi-scale GRU (Figure 1) is composed of three GRUs at multiple scales. The first is 1/32 the resolution of the disparity map and the output is upsampled and imputed into the higher-resolution 1/16 GRU. Then, the output is upsampled and inputted into the highest resolution 1/8 GRU to output the update to the disparity map estimate to obtain the new disparity map update with an element-wise addition. The correlation-pyramid lookup is used to input into the highest resolution 1/8 GRU and the downsampled current disparity map estimate is used as an input for the other two GRUs, and the middle resolution 1/16 GRU also uses the highest resolution 1/8 feature map as an input.

For training the network, the loss is computed after each update to the disparity map prediction with disparity predictions that took more update steps having a greater weighting.

#### Performance Evaluation

This method was tested on the KITTI-2015 stereo test dataset [36]. This dataset contains 200 scenes captured from a stereo camera setup mounted on a car. The ground truth is captured using a LiDAR sensor and the evaluation is performed on the percentage of the erroneous pixels averaged over all 200 scenes. The predicted disparity map values represent the pixel disparity between the left–right image so erroneous pixels are where the absolute difference between the ground truth and prediction is greater than a pixel threshold which is chosen to be 3 pixels. The method resulted in 1.96% for all pixels, 2.89% for foreground pixels and 1.75% for background pixels.

The qualitative results in Figure 2 show the fine depth detail in the fence and sign posts. This is due to it outputting the image at full resolution without the need to upsample from a lower-resolution prediction.

### 2.2. Practical Stereo Matching via Cascaded Recurrent Network with Adaptive Correlation

This method, CREStereo [27], computes a disparity map from a pair of stereo images using a deep convolutional neural network. The disparity map is computed using a recurrent-based architecture to refine the disparity prediction starting from a zero-initialised disparity map.

First, a feature network is used on both left–right stereo images to extract the feature maps which are used to construct three-level feature pyramids for both of the input images. Next, a positional encoding is added to the feature pyramid and then it is inputted through a self-attention layer to obtain the final feature pyramids which are used as the inputs for the recurrent update module.

Next, the recurrent update module uses an adaptive group correlation layer similar to [2]. This layer outputs the inputs for the recurrent neural network from the feature pyramids. This is performed by first using a cross-attention layer to obtain the grouped features which are then sampled using the current predicted disparity map. The sampling is conducted using a search window using fixed offsets which alternates between a 1D and 2D search to allow for stereo matching that does not lie on the epipolar line. The feature pyramid is also used to compute the learned offsets to allow for a deformable search window to be used by extending the method for deformable convolutions [37]. The pyramid sampling will collect features in a window to be used to construct a correlation volume of size H×W×D where H,W is the input image height and width, and *D* is the number of correlation pairs which is smaller than *W*. The correlation at x,y,d in the correlation volume is computed as follows:(1)$$\begin{array}{c} corr\left(x,y,d\right)=\frac{1}{c}\sum_{i=1}^{C}F_{1}\left(i,x,y\right)F_{2}\left(i,x^{\prime \prime },y^{\prime \prime }\right) \end{array}$$
where $x^{\prime \prime }=x+f\left(d\right)+dx$, $y^{\prime \prime }=y+g\left(d\right)+dy$. $F_{1},F_{2}$ are the feature maps, f(d),g(d) are the fixed offsets which alternate between 1D when g(d)=1 and 2D when g(d)∈[−4,4] with f(d)∈[−4,4] for both and dx,dy are the learned offset which are computed from the feature pyramids.

The correlation volume is used as an input to the recurrent neural network which is formed of multiple gated recurrent units (GRU). The GRUs are each at different scales, and then the final disparity map is computed using cascaded refinement where the outputted disparity map from a lower-level GRU is upsampled and then used as the initial disparity map for the next GRU level to finally obtain the highest level GRU output which is used as the final disparity map.

#### Performance Evaluation

This method was tested on the KITTI-2015 stereo test dataset using the percentage of erroneous pixels with a three-pixel threshold. The method resulted in 1.69% for all pixels, 2.86% for foreground pixels and 1.45% for background pixels.

The qualitative results in Figure 3 show fine detail in the fence and sign posts similar to [26] because it can also output the full resolution.

### 2.3. Hierarchical Neural Architecture Search for Deep Stereo Matching

This method, LEAStereo (Learning Effective Architecture Stereo) [28], computes a disparity map from only two RGB stereo images using a deep convolutional neural network. The network architecture is found using an NAS (Neural Architecture Search) and then the found network architecture is trained to obtain the final model.

The neural network has four stages to compute the disparity map from the stereo image pair. The first stage uses a 2D feature network to extract a pair of feature maps from the stereo images. The second stage combines the feature maps into a 4D feature volume. The third stage is a 3D matching network that will compute a matching cost for each possible disparity using the 4D feature volume as the input. The final stage will project the 3D cost volume to a 2D disparity map using a soft-argmin layer. Only the feature network and the matching network have trainable parameters so an NAS is used to find the best architecture for these networks.

The architecture search is conducted using a differentiable architecture search [38] where there is a continuous relaxation of the discrete architectures, so the best architecture can be found using gradient descent.

#### 2.3.1. Cell-Level Search Space

The feature network and the matching network are both deep convolutional neural networks that are constructed with computational cells which are stacked in layers and share the same structure. The architecture search is conducted using a two-level hierarchical search [39] that will find the best cell-level and network-level structures.

The cells (Figure 4) are direct acyclic graphs that consist of an ordered sequence of *n* nodes. The nodes are the feature maps in the convolutional network and the edges are operations that transform the feature maps; so, a node is computed by applying the operations and then combining the feature maps with an element-wise addition. The first two nodes are used as inputs, so for cell $C_{l}$ in layer *l*, the two input nodes are outputs from the two preceding cells ($C_{l-2},C_{l-1}$). The cell output is computed by concatenating nodes that are not input nodes to the cell. The cells in the final model that will be found will only have two input connections for each node. The architecture search will find these connections and the operation.

To make the architecture continuous in the search phase, the nodes have all the possible connections, so node *i* is connected to nodes {1,2,...,i−1}, and the edges have all the possible operations from the set of candidate operations. In the search phase, the output of a node is computed as follows:(2)$$\begin{array}{c} s^{\left(j\right)}=\sum_{i⇝j}o^{\left(i,j\right)}\left(s^{\left(i\right)}\right). \end{array}$$
where ⇝ denotes node *i* is connected to node *j*. $o^{\left(i,j\right)}$ is defined as follows:(3)$$\begin{array}{c} o^{\left(i,j\right)}\left(x\right)=\sum_{r=1}^{v}\frac{\exp(\alpha_{r}^{\left(i,j\right)})}{\sum_{s=1}^{v}\exp\left(\alpha_{s}^{\left(i,j\right)}\right)}o_{r}^{\left(i,j\right)}\left(x\right) \end{array}$$
where $o_{r}^{\left(i,j\right)}\left(x\right)$ is the r-th operation between nodes *i* and *j*. $\alpha^{\left(i,j\right)}=\left(\alpha_{1}^{i,j},\alpha_{2}^{i,j},...,\alpha_{v}^{i,j}\right)$ is a weight-mixing vector that is used in the softmax function to weight the operations. After the search phase, the operation to be used $O_{r^{*}}^{\left(i,j\right)}$ is selected as the operation with the highest corresponding weight after the search phase $r^{*}=\arg\max_{r}\alpha_{r}^{\left(i,j\right)}$. The set of candidate operations for the feature net is $O^{F}=\left\{“3\times 3\;\mathrm{convolution}”,“\mathrm{skip}\;\mathrm{connection}”\right\}$. The set of candidate operations for the matching net is $O^{M}=\left\{“3\times 3\times 3\;\mathrm{convolution}”,“\mathrm{skip}\;\mathrm{connection}”\right\}$. The top two highest operations are selected for each node, so each node has two input connections. Following ResNet [40], the cells are modified to include a residual connection between the output of the previous cell and the cell’s output.

#### 2.3.2. Network-Level Search Space

The network-level search space will have a fixed number of layers *L* that will allow the network to downsample, upsample or keep the same spatial resolution of the feature maps at each layer. The search space has a maximum and minimum resolution the feature map can be, and the network will downsample by a scale of 1/2 and upsample by a scale of 2.

To make the architecture continuous at the network level, all possible configurations are computed with weighting parameters. In the search phase, the output $h_{\left(s\right)}^{\left(l\right)}$ of layer *l* at spatial resolution *s* is computed as follows:(4)$$\begin{array}{c} H_{\left(s\right)}^{\left(l\right)}={\bar{\beta }}_{\left(s/2\right)}Cell\left(h_{\left(s/2\right)}^{\left(l-1\right)},h_{\left(s\right)}^{\left(l-2\right)}\right)\\ +{\bar{\beta }}_{\left(s\right)}Cell\left(h_{\left(s\right)}^{\left(l-1\right)},h_{\left(s\right)}^{\left(l-2\right)}\right)\\ +{\bar{\beta }}_{\left(2s\right)}Cell\left(h_{\left(2s\right)}^{\left(l-1\right)},h_{\left(s\right)}^{\left(l-2\right)}\right) \end{array}$$
where $Cell(h^{\left(l-1\right)},h^{\left(l-2\right)})$ is the cell computation which takes the outputs from the previous two layers (l−1,l−2) as inputs. ${\bar{\beta }}_{s}$ is the weight parameter after the softmax function has been applied. The best network architecture is selected as the highest weights for each layer.

The network used 6 layers for the feature network and 12 layers for the matching network. A maximum resolution of 1/3 of the input size and a minimum of 1/24 were used for the search space. The cells used five nodes.

#### 2.3.3. Optimisation

The network is searched and trained in an end-to-end manner, so the loss function uses the final disparity output computed by the model and the ground-truth disparity. The loss function is defined as follows:(5)$$\begin{array}{c} L=\ell \left(d_{pred}-d_{gt}\right),\mathrm{where}\;\ell \left(x\right)=\left\{\begin{array}{cc} 0.5x^{2}, & \;\mathrm{if}\;|x|<0\\ |x|-0.5, & \;\mathrm{otherwise}\; \end{array}\right. \end{array}$$
where $d_{pred}$ is the predicted disparity map by the network and $d_{gt}$ is the ground-truth disparity map.

The training is performed with two disjoint training sets; one optimises the network parameters and the other optimises the architecture search parameter α,β. The optimisation is performed by alternating the update of the network weights and the network search parameters. The network has been trained on the SceneFlow dataset [13].

#### 2.3.4. Performance Evaluation

This method was tested on the KITTI-2015 stereo test dataset using the percentage of erroneous pixels with a three-pixel threshold. The method resulted in 1.65% for all pixels, 2.91% for foreground pixels and 1.40% for background pixels.

The qualitative results in Figure 5 show less fine detail with smoother edges. This is due to it outputting the disparity map at a lower resolution and then upsampling it.

### 2.4. PENet: Towards Precise and Efficient Image-Guided Depth Completion

This method, PENet [29], computes a depth map from a sparse LiDAR depth map and a single RGB image. The method uses a deep convolutional neural network to compute an initial depth map which is then refined with a convolutional spatial propagation network.

The neural network for computing the initial depth map uses a two-branch backbone as shown in Figure 6. Both of the branches will output a dense depth map prediction and confidence weight map which are used to fuse into a single depth map prediction.

The first branch is the colour-dominant branch which uses both the RGB image and the sparse depth map as inputs. The network uses an encoder–decoder architecture with symmetric skip connections. This network will output a colour-dominant depth map prediction and a confidence map which has confidence values for each pixel.

The second branch is the depth-dominant branch which uses the output depth map prediction of the colour-dominant branch and the sparse depth map for the network inputs, similar to [41]. This branch has a similar architecture as the colour-dominant branch and it also takes the features in the colour-dominant decoder as extra inputs in the corresponding layers for the depth-dominant encoder.

In both the network branches, a geometric convolutional layer is used which is similar to a standard convolutional layer, but it uses additional 3D position maps which are concatenated to the feature inputs to form the input to the geometric convolutional layer. The position map is derived from the sparse LiDAR depth map *D* as follows:(6)$$\begin{array}{c} Z=D,X=\frac{(u-u_{0})Z}{f_{x}},Y=\frac{(v-v_{0})Z}{f_{y}} \end{array}$$
where (u,v) are the coordinates of a pixel and $u_{0},v_{0},f_{x},f_{y}$ are the intrinsic parameters of the camera.

The initial depth map prediction is then refined using a convolutional spatial propagation network [14] with dilated convolutions [42].

The depth refinement is conducted in iterations using affinity weight maps learnt by the network. For each pixel, it aggregates information propagated from pixels within the neighbourhood for the initial depth map $D_{0}$ that has been produced by the network as follows:(7)$$\begin{array}{c} D_{i}^{t+1}=W_{ii}D_{i}^{0}+\sum_{j\in N(i)}W_{ji}D_{j}^{t} \end{array}$$
where $W_{ij}$ is the affinity between pixel *i* and pixel *j*. The network will learn the affinity maps.

The $l_{2}$ loss is used for training which is defined as follows:(8)$$\begin{array}{c} L\left(\hat{D}\right)=\left|\right|\left(\hat{D}-D_{gt}\right)⊙1\left(D_{gt}>0\right){\left|\right|}^{2} \end{array}$$
where $\hat{D}$ is the final predicted depth map, $D_{gt}$ is the ground truth, 1 is an indicator and ⊙ is an element-wise multiplication. This is so it only uses the valid pixels in the ground truth for training. For early epochs, a loss is applied to the intermediate depth predictions from the colour-dominant branch and the depth-dominant branch as follows:(9)$$\begin{array}{c} L=L\left(\hat{D}\right)+\lambda_{cd}L\left({\hat{D}}_{cd}\right)+\lambda_{dd}L\left({\hat{D}}_{dd}\right) \end{array}$$
where $\lambda_{cd}$ and $\lambda_{dd}$ are hyperparameters.

#### Performance Evaluation

This method was tested on the KITTI depth completion validation dataset. This dataset contains 200 scenes captured from a LiDAR sensor and a stereo camera setup mounted on a car. The ground truth is captured using the LiDAR sensor, accumulated from multiple frames. The depth is measured in metres in the data and scaled to millimetres for the evaluation which is performed with the mean absolute error (MAE) in mm, the root mean squared error (RMSE) in mm, the inverse mean absolute error (iMAE) in 1/km and the inverse root mean squared error (iRMSE) in 1/km. The method resulted in 209.00 for the MAE, 757.20 for the RMSE, 0.92 for the iMAE and 2.22 for the iRMSE.

The qualitative results in Figure 7 show noise in the disparity maps because the sparse LiDAR data have large areas with no depth information so there will be high uncertainty in those regions.

### 2.5. SemAttNet: Towards Attention-Based Semantic-Aware-Guided Depth Completion

This method, SemAttNet [30], computes a depth map using a sparse LiDAR depth map and a single RGB image as inputs (and a semantic map). This method first uses a convolutional neural network to output an initial coarse depth map which is refined with a spatial propagation network.

The network used for computing the initial depth map uses a three-branch backbone which consists of a colour branch, a depth branch and a semantic branch as shown in Figure 8. The depth branch takes the output depth map predictions computed by the colour and semantic branches as well as the sparse LiDAR depth as inputs.

The network uses a semantic-aware multi-modal attention-based fusion block in the initial network [43]. It is first used to fuse the feature maps from the colour and semantic branches and then it is used in the depth branch to fuse the features from the other two branches.

The colour branch uses an encoder–decoder network with skip connections which takes the colour image and the sparse depth map as inputs and will output a dense depth map and a confidence map.

The semantic branch takes the depth map computed with the colour-guided branch as an input as well as the sparse depth map and the semantic image, similar to [29,41]. The network will output a dense depth map and a confidence map. The network uses a similar encoder–decoder architecture and uses the decoder features from the colour-guided branch and fuses them with the features in the encoder network. The semantic image is computed from the RGB image with a pretrained WideResNet38 [44] model.

The depth branch uses the outputted depth maps from the colour branch and the semantic branch as inputs as well as the sparse depth map. The branch uses a similar encoder–decoder network architecture to the other branches, with the decoder features from both branches being fused with the features in the encoder network.

The decoder features are fused with the encoder features using a semantic-aware multi-modal attention-based fusion block. This uses channel-wise attention and then spatial-wise attention to output a single feature map.

The final depth maps outputted from each branch are fused using the corresponding learnt confidence maps outputted by each branch, similar to [29,45].

The training loss computes the $l_{2}$ loss for each branch-outputted depth map and the final fused depth map using the ground-truth depth map.

Then, the fused depth map from the three-branch network is refined using the CSPN++ [14] with dilated convolutions.

#### Performance Evaluation

This method was tested on the KITTI depth completion validation dataset. The method resulted in 205.49 for the MAE, 709.41 for the RMSE, 0.90 for the iMAE and 2.03 for the iRMSE.

The qualitative results in Figure 9 show that, similar to Figure 7, the disparity map has noise in the regions where there are no LiDAR data.

### 2.6. Dynamic Spatial Propagation Network for Depth Completion

This method, DySPN [31], computes a depth map from a sparse LiDAR depth map with a single RGB image as well for guidance. A coarse dense depth map is initially computed with a convolutional neural network and then the coarse depth map is refined using a dynamic spatial propagation network.

The initial coarse depth map is computed from the sparse LiDAR depth map using a Unet-like [46] encoder–decoder network with ResNet-34 [40] as the backbone. This network will output attention maps, the initial depth map, an affinity matrix and a confidence prediction for the input depth [14,29,47] as shown in Figure 10.

The depth map is updated with adaptive affinity weights to refine the depth map over *N* steps. This can be defined as $V^{t+1}=G^{t}V^{t}$ for t∈{0,1,2,…,N} with the depth map reshaped to the one-dimensional vector $V^{t}\in R^{mn}$, and the global adaptive affinity matrix $G^{t}\in R^{mn\times mn}$ which contains all adaptive affinity weights can be defined as follows:(10)$$\begin{array}{c} G^{t}=\left[\begin{array}{cccc} 1-{\tilde{\lambda }}_{0,0}^{t} & {\tilde{w}}_{0,0}^{t}\left(1,0\right) & \cdots & {\tilde{w}}_{0,0}^{t}\left(m,n\right)\\ {\tilde{w}}_{1,0}^{t}\left(1,0\right) & 1-{\tilde{\lambda }}_{1,0}^{t} & \cdots & {\tilde{w}}_{1,0}^{t}\left(m,n\right)\\ \vdots & \vdots & \ddots & \vdots \\ {\tilde{w}}_{m,n}^{t}\left(1,0\right) & {\tilde{w}}_{m,n}^{t}\left(1,0\right) & \cdots & 1-{\tilde{\lambda }}_{m,n}^{t} \end{array}\right] \end{array}$$
where ${\tilde{w}}_{i,j}^{t}\left(a,b\right)=\pi_{i,j}^{t}\left(a,b\right)w_{i,j}\left(a,b\right)$ is the adaptive affinity weight which is computed using the fixed affinity weights $w_{i,j}\left(a,b\right)$ and the global attention $\pi_{i,j}^{t}\left(a,b\right)$. ${\tilde{\lambda }}_{0,0}^{t}=\sum_{a\neq i,b\neq j}$ so that each row will sum to 1.

The DySPN reduces the computational complexity of the global adaptive affinity update by only computing for a neighbourhood. The affinity weights for the neighbours of the same distance are set to be equal and the neighbourhood is also deformable [48].

The $L_{1}$ and $L_{2}$ loss are both used to compute the training loss which is defined as $Loss\left(h^{gt},h^{N}\right)=L_{1}\left(h^{gt},h^{N}\right)+L_{2}\left(h^{gt},h^{N}\right)$ where $h^{N}$ is the depth map after *N* steps and $h^{gt}$ is the ground-truth depth map.

#### Performance Evaluation

This method was tested on the KITTI depth completion validation dataset. The method resulted in 192.71 for the MAE, 709.12 for the RMSE, 0.82 for the iMAE and 1.88 for the iRMSE.

The qualitative results in Figure 11 also show similar noise in the depth map to the other depth completion methods.

### 2.7. Volumetric Propagation Network: Stereo-LiDAR Fusion for Long-Range Depth Estimation

This method [32] computes a depth map using the RGB stereo images and a LiDAR point cloud of the scene. The method pipeline is shown in Figure 12. First, a fusion volume is constructed to represent the scene as a three-dimensional voxel grid, evenly distributed along the depth range. The fusion volume is filled with information from the stereo images, first by using a feature extraction network and then converting the voxel coordinates onto the stereo images using the calibration parameters. Next, the point clouds are also converted into the fusion volume and stored as a binary occupancy representation.

The final information embedded into the fusion volume is the point cloud features which are extracted using FusionConv layers that cluster neighbouring points from the point cloud that lie within a voxel window. These points are fused with the corresponding features in the left feature map and then the weighted geometric distance is computed to obtain the weights for the convolution operation on the neighbouring point features.

The final fusion volume is fed through a stacked hourglass network [1,49] that consists of multiple encoder and decoder networks which compute multiple cost–volume estimates. The depth is computed using the soft-min function. The training total loss is computed with a weighted sum of the loss for each depth map estimate in the hourglass network.

#### Performance Evaluation

This method was tested on the KITTI depth completion validation dataset. The method resulted in 205.1 for the MAE, 636.2 for the RMSE, 0.9870 for the iMAE and 1.8721 for the iRMSE.

The qualitative results in Figure 13 show the depth map has less noise than the depth completion method because it also uses stereo matching.

### 2.8. Three-Dimensional LiDAR and Stereo Fusion using Stereo Matching Network with Conditional Cost Volume Normalization

This method [33] computes a disparity map by using the RGB images and sparse disparity maps of a LiDAR point cloud projected onto the RGB image views. The stereo and LiDAR data are fused by concatenating the RGB image with the LiDAR disparity map to create a four-channel image and then this is inputted into a single feature extraction network using 2D convolutional layers to obtain the feature maps for both the left and right images. The features are used to construct a 4D feature volume which contains all the potential matches. Next, a matching network is used to apply regularisation by using 3D convolutional layers to obtain the final cost volume. Finally, the soft-min function can be used to obtain the final disparity map.

The cost–volume is normalised by using Conditional Cost–Volume Normalisation which is inspired by Conditional Batch Normalisation [50,51]. In conditional Batch Normalisation, the learnable parameters are defined as functions of conditional information, and for conditional cost–volume normalisation, the information used is the pixels in the LiDAR data. For the pixels with no values, a learnable constant value is used instead. The conditional cost–volume method is approximated with a hierarchical extension to reduce the number of parameters by computing an intermediate vector conditioned on each LiDAR pixel and then modulating this with learnable parameters for each depth in the volume. This conditional cost–volume regularisation is used in the matching network to compute the final cost–volume from the feature volume.

#### Performance Evaluation

This method was tested on the KITTI depth completion validation dataset. The method resulted in 252.5 for the MAE, 749.3 for the RMSE, 0.8069 for the iMAE and 1.3968 for the iRMSE.

The qualitative results in Figure 14 show less noise than the depth completion methods; however, the visual quality is lower than the methods that only use stereo data.

### 2.9. Noise-Aware Unsupervised Deep LiDAR–Stereo Fusion

This method [34] fuses a pair of stereo RGB images with the LiDAR-projected disparity maps using an unsupervised training method without the need for training data with a ground-truth disparity map.

The method uses a deep convolutional neural network that computes the final disparity maps using multiple stages, similar to [7] as shown in Figure 15. The first stage extracts the feature from the stereo images and features from the LiDAR disparity maps. The network uses sparsity invariant convolutional layers [52] for the sparse LiDAR disparity maps. The stereo and LiDAR feature maps are concatenated into a single feature map for both the left and right Images. The next stage constructs a 4D feature volume using both feature maps and then feature matching is computed using 3D convolutions to compute the cost–volume. Finally, the disparity map is computed using a soft-argmin on the cost–volume.

The data are passed through this network two times. The first produces the initial disparity maps that are used to remove the noisy LiDAR disparities so the filtered LiDAR disparity maps are used in the second pass to compute the final disparity map. The network is trained using an unsupervised loss function that is the sum of an image warping loss, a LiDAR loss, a smoothness loss and a plane fitting loss. The image warping loss computes the photometric consistency between the stereo images using the final disparity map. The LiDAR loss computes the distance between the filtered LiDAR disparity map and the final disparity map. The smoothness loss computes the smoothness of the final disparity map. The plane fitting loss computes the distance between the final disparity map and a disparity map created by projecting the disparity values onto a local plane. The network is only updated after the LiDAR disparities have been filtered. Backpropagation will only go through the final network pass after filtering.

#### Performance Evaluation

This method was tested on the KITTI stereo dataset with only the frames that have LiDAR data available to create a subset of 141 frames (KITTI-141 subset). The evaluation was performed with the absolute relative error (Abs rel), the maximum ratio threshold error (δ<1.25) and the percentage of erroneous pixels, with pixel thresholds of 2, 3 and 5 (>2 px, >3 px, >5 px). The method resulted in 0.0350 for the Abs rel, 0.0287 for >2 px, 0.0198 for >3 px, 0.0126 for >3 px and 0.9872 for δ<1.25.

The qualitative results in Figure 16 show less fine detail in the disparity map than [26,27] but less noise than the depth completion methods.

## 3. Comparative Evaluation of the State-of-the-Art Fusion Techniques

In this section, we compare the performance of the multi-modal fusion methods by using the results from the KITTI depth completion dataset and the KITTI stereo dataset.

We compare the Volumetric Propagation Network (VPN) method, the 3D LiDAR and Stereo Fusion using a Stereo Matching Network with Conditional Cost–Volume Normalization (CCVN) method, the PENet: Towards Precise and Efficient Image-Guided Depth Completion (PENet) method, the SemAttNet: Towards Attention-based Semantic-Aware-Guided Depth Completion (SemAttNet) method and the Dynamic Spatial Propagation Network for Depth Completion (DySPN).

We compare the Raft-stereo: Multilevel recurrent field transforms for the stereo matching (RAFT) method, the Noise-Aware Unsupervised Deep LiDAR–Stereo Fusion (LiDARStereoNet) method and the Hierarchical Neural Architecture Search for Deep Stereo Matching (LEAStereo) method on the KITTI stereo dataset.

The compared results (Table 2) show the methods tested on the KITTI depth completion dataset which has LiDAR and stereo data. The other results (Table 3) show the methods tested on the KITTI stereo dataset and evaluated using a pixel threshold error which is the percentage of values with a greater error than 3 pixels. The LEAStereo method has the highest accuracy on the stereo dataset, using only a pair of RGB stereo images. The VPN method has the highest accuracy with the RMSE metric and the CCVN method with the iMAE and iRMSE metrics, and these are evaluated on the depth completion dataset which has LiDAR and RGB so the accuracy will be higher than RGB only. However, the DySPN method has the highest accuracy with the MAE metric which uses LiDAR and only one RGB image.

The compared methods (Table 4) show the strengths and limitations of each method. The methods with a slower inference time take over 1 s to compute the depth map due to using stereo and LiDAR data as the input. The methods with a very fast inference time take less than 0.25 s to compute the depth map due to using LiDAR and RGB without stereo data. The methods that have a high memory usage will be limited by the memory available when computing the depth maps so downsampling is used to reduce the memory usage which can reduce the accuracy.

## 4. Results Analysis, Discussions and Application Areas

The methods that we compare are all evaluated using the KITTI dataset [36] which is developed for the application of autonomous driving. The dataset was created using sensors mounted to a vehicle and contains diverse scenes of urban, rural and highways.

Shown in Table 4, the methods VPN, CCVN, LiDARStereoNet and LEAStereo are limited by the high memory usage because they all use a feature volume to compute the final depth map which has a high memory cost. This can limit the methods to outputting a depth map at a lower resolution and then upsampling it, which will result in losing fine 3D detail. These methods perform well on the KITTI dataset, with VPN having the highest accuracy and MAE and CCVN having the highest accuracy for the iMAE and IRMSE. LEAStereo also has the highest accuracy on the KITTI stereo dataset; however, these methods would not be able to efficiently compute depth maps for datasets with larger resolution data.

The methods RAFT and CREStereo use a recurrent architecture that indexes features to improve the computational and memory efficiency, so increasing the resolution of the RGB input images would not have as high a memory cost as the feature volume methods.

The methods PENet, SemAttNet and DySPN do not use stereo RGB images as inputs; instead, they only use a single RGB image which reduces the memory cost of the computation and would be more efficient at computing depth maps for higher resolution inputs.

The specific area these methods can be applied to is computing a 3D model of a road environment to be used for autonomous driving. The methods RAFT, CREStereo and LEAStereo all rely on feature matching from stereo images so they would only be suitable for autonomous driving in environments with texture so features can be easily matched. These methods would perform worse for autonomous driving in environments with texture-less areas, such as building interiors with walls without any patterns. The methods using the LiDAR depth map as an input, such as the PENet, SemAttNet, DySPN, VPN, CCVN, and LiDARStereoNet, would be more suitable for these indoor environments.

## 5. Conclusions

The state-of-the-art LiDAR and stereo fusion methods show a lower accuracy than the current state-of-the-art depth completion or stereo methods when using the MAE or pixel threshold error to evaluate accuracy. In the other metrics used (RMSE, iMAE and iRMSE), the LiDAR–stereo fusion methods show a higher accuracy. The LiDAR–stereo methods also require more computing time and have higher memory usage when compared to the LiDAR–RGB methods. The LiDAR–stereo methods would be more suitable in environments where one of the sensor modalities will have a low accuracy.

## Figures and Tables

**Figure 1 sensors-22-09364-f001:**
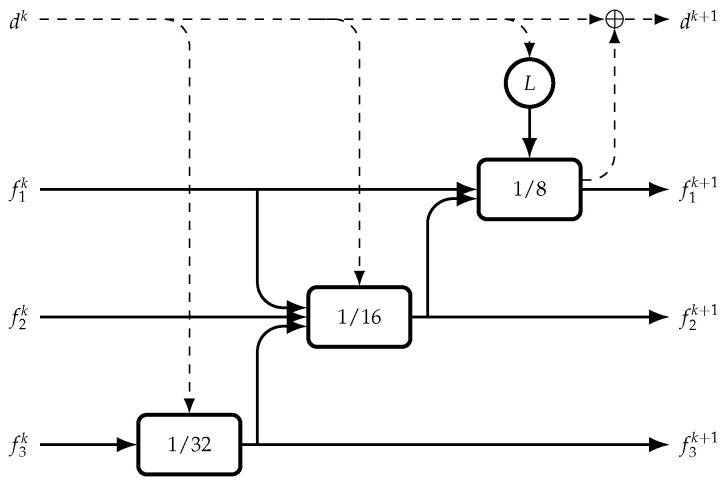
The multi-scale GRU shown with 3 feature maps $f_{1},f_{2},f_{3}$ at different scales 1/8,1/16,1/32. The disparity map $d^{k}$ shown is updated by adding the output of the lowest scale GRU 1/8, the correlation-pyramid lookup operation is shown as *L* and the result is inputted in the lowest scale GRU 1/8. Downsampling and upsampling are used to change the scales of the feature maps or disparity map when being passed between the GRUs.

**Figure 2 sensors-22-09364-f002:**
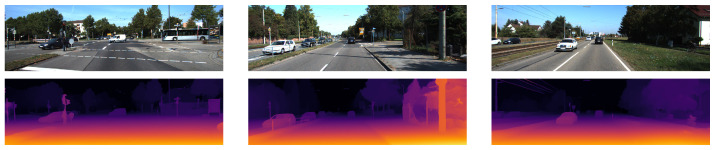
The qualitative results from [26] show the left RGB images from 3 scenes in the KITTI-2015 stereo dataset in the top row and the corresponding disparity map predictions in the bottom row are shown using colour to represent the disparity values.

**Figure 3 sensors-22-09364-f003:**
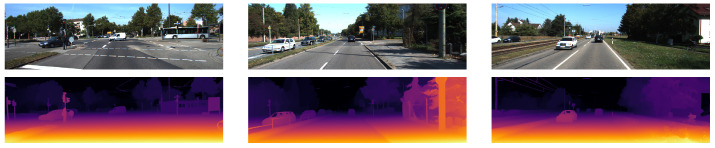
The qualitative results from [27] show the left RGB images from 3 scenes in the KITTI-2015 stereo dataset in the top row and the corresponding disparity map predictions in the bottom row are shown using colour to represent the disparity values.

**Figure 4 sensors-22-09364-f004:**
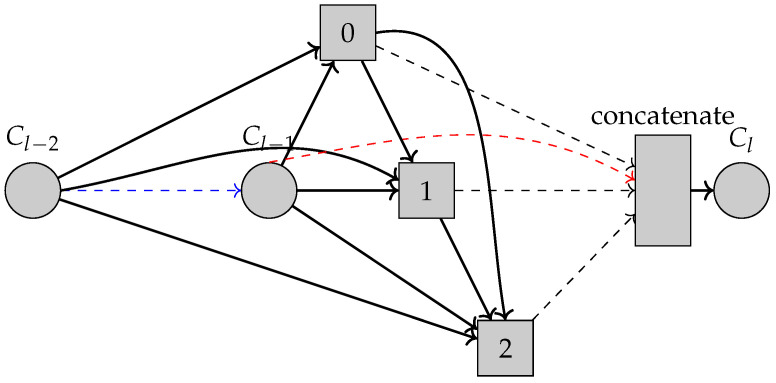
Cell-level search space. The cell output $C_{l}$ is computed using the previous 2 cell outputs $C_{l-2},C_{l-1}$. The intermediate nodes are shown as 0,1,2, and for each input connection to these nodes, an operation will be searched for. The red dashed arrow shows the residual connection and the blue dashed arrow shows the previous cell that uses $C_{l-2}$ as one of its inputs to compute $C_{l-1}$. The intermediate nodes are concatenated, shown with black dashed arrows to obtain the final output $C_{l}$.

**Figure 5 sensors-22-09364-f005:**
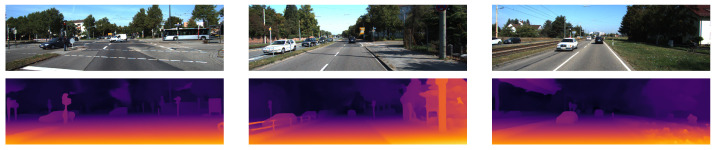
The qualitative results from [28] show the left RGB images from 3 scenes in the KITTI-2015 stereo dataset in the top row and the corresponding disparity map predictions in the bottom row are shown using colour to represent the disparity values.

**Figure 6 sensors-22-09364-f006:**
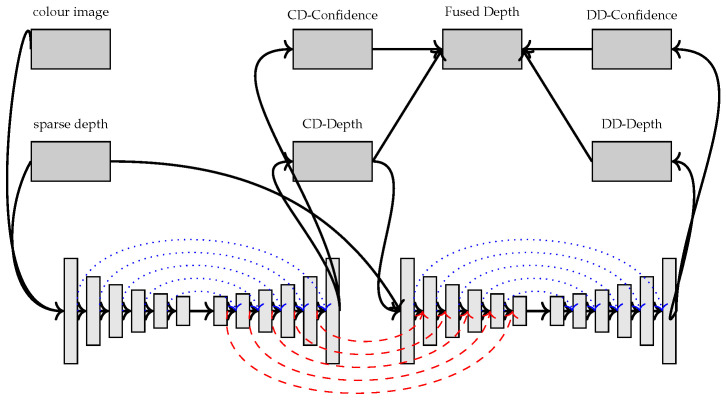
The inputs shown are the colour image and the sparse depth which are inputted into the network to compute the fused depth map which is then further refined. The blue connections show where the features are combined with addition and the red connection show where they are combined with concatenation.

**Figure 7 sensors-22-09364-f007:**
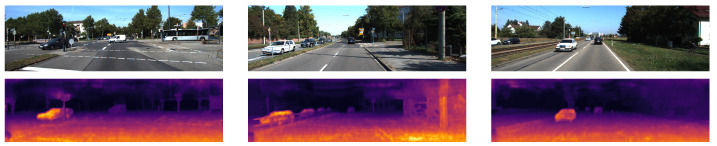
The qualitative results from [29] show the left RGB images from 3 scenes in the KITTI-2015 stereo dataset in the top row and the corresponding disparity map predictions in the bottom row are shown using colour to represent the disparity values. The raw dataset was used for the LiDAR data mapped onto the scenes in the stereo dataset and then the results were converted from depth values to disparity using the calibration parameters.

**Figure 8 sensors-22-09364-f008:**
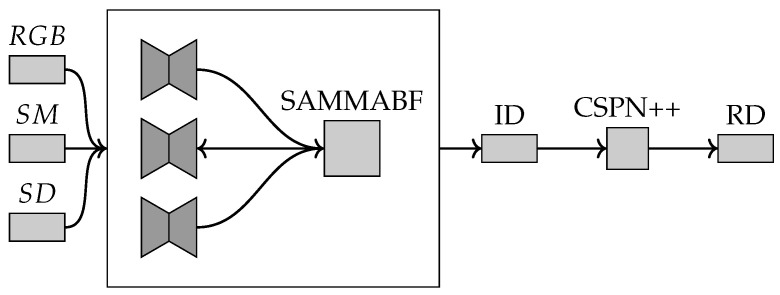
The inputs shown on the left are the RGB image, the semantic map computed from the RGB image and the sparse depth captured by the LiDAR sensor. The features from the three-branch backbone are fused using the semantic-aware multi-modal attention-based fusion block (SAMMABF) and will output the initial depth map shown as ID. This is refined with the CSPN++ [14] with dilated convolutions [42] to output the refined depth map shown as RD.

**Figure 9 sensors-22-09364-f009:**
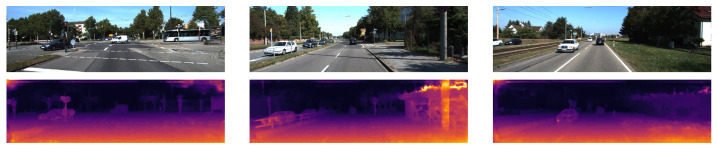
The qualitative results from [30] show the left RGB images from 3 scenes in the KITTI-2015 stereo dataset in the top row and the corresponding disparity map predictions in the bottom row are shown using colour to represent the disparity values. The raw dataset was used for the LiDAR data mapped onto the scenes in the stereo dataset and then the results were converted from depth values to disparity using the calibration parameters.

**Figure 10 sensors-22-09364-f010:**
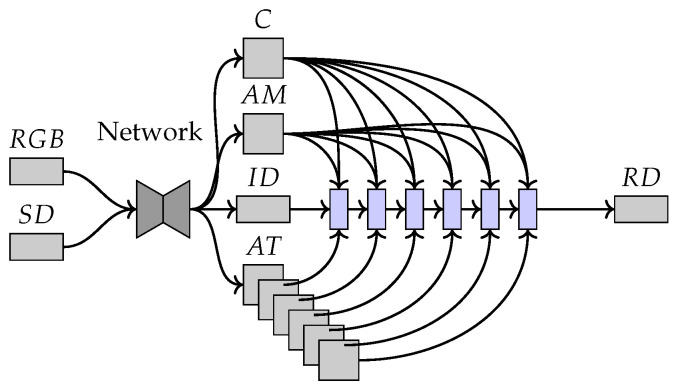
The inputs shown on the left are the RGB image and the sparse depth captured by the LiDAR sensor. The encoder–decoder network outputs the confidence shown as *C*, the affinity matrix as AM, the initial depth prediction as ID and the attention maps as AT. The initial depth is refined over 6 steps of the DySPN, shown in blue, to obtain the final refined depth map shown as RD.

**Figure 11 sensors-22-09364-f011:**
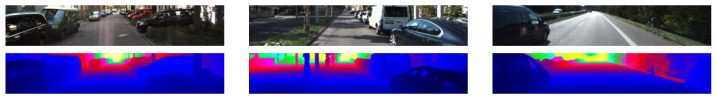
The qualitative results from [31] show the left RGB images from 3 scenes in the KITTI depth completion dataset in the top row and the corresponding depth map predictions in the bottom row are shown using colour to represent the depth values.

**Figure 12 sensors-22-09364-f012:**
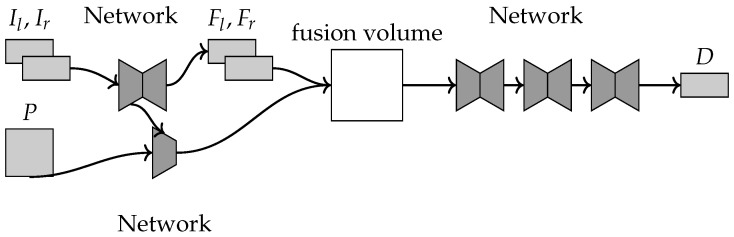
The inputs shown are the point cloud P and the left–right RGB image pair $I_{l},I_{r}$ used to compute feature maps $F_{l},F_{r}$, and the final depth map D is computed from the fusion volume.

**Figure 13 sensors-22-09364-f013:**
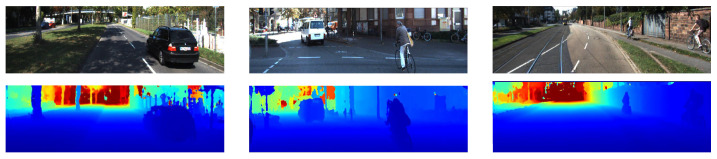
The qualitative results from [32] show the left RGB images from 3 scenes in the KITTI raw dataset in the top row and the corresponding depth map predictions in the bottom row are shown using colour to represent the depth values.

**Figure 14 sensors-22-09364-f014:**
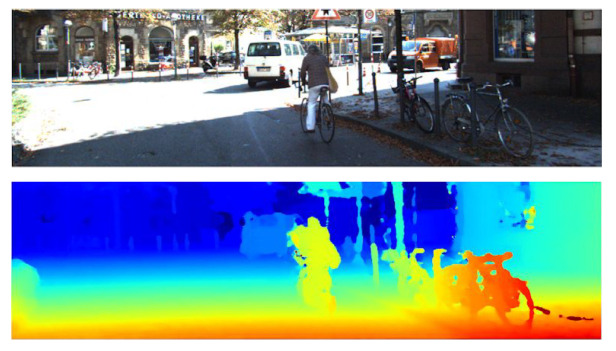
The qualitative results from [33] show the left RGB image from a scene in the KITTI depth completion dataset in the top row and the corresponding depth map prediction in the bottom row are shown using colour to represent the depth values.

**Figure 15 sensors-22-09364-f015:**
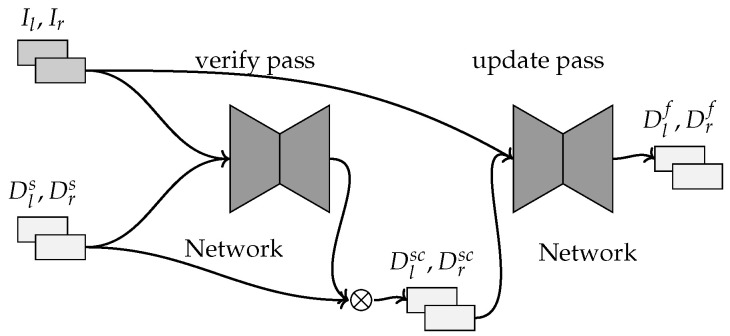
The inputs shown are the left–right RGB images $I_{l},I_{r}$ and the left–right sparse disparity maps $D_{l}^{s},D_{r}^{s}$. After the verify pass, the filtered disparity maps are computed $D_{l}^{sc},D_{r}^{sc}$ and this removes disparities that are not consistent. After the update pass, the fused disparity maps are computed $D_{l}^{f},D_{r}^{f}$.

**Figure 16 sensors-22-09364-f016:**
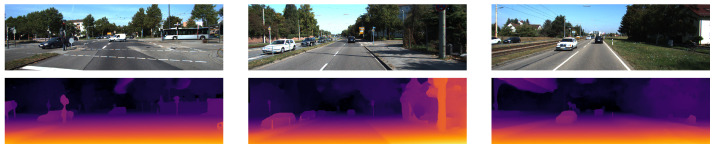
The qualitative results from [34] show the left RGB images from 3 scenes in the KITTI-2015 stereo dataset in the top row and the corresponding disparity map predictions in the bottom row are shown using colour to represent the disparity values. The raw dataset was used for the LiDAR data mapped onto the scenes in the stereo dataset and then the results were converted from depth values to disparity using the calibration parameters.

**Table 1 sensors-22-09364-t001:** The methods categorised by the input data used.

Stereo Methods	LiDAR and Stereo Methods	LiDAR and RGB Methods
Raft-stereo: Multilevel recurrent field transforms for stereo matching [26]. Practical Stereo Matching via Cascaded Recurrent Network with Adaptive Correlation [27]. Hierarchical Neural Architecture Search for Deep Stereo Matching [28].	PENet: Towards Precise and Efficient Image Guided Depth Completion [29]. SemAttNet: Towards Attention-based Semantic Aware Guided Depth Completion [30]. Dynamic Spatial Propagation Network for Depth Completion [31].	Volumetric Propagation Network: Stereo-LiDAR Fusion for Long-Range Depth Estimation [32]. 3D LiDAR and Stereo Fusion using Stereo Matching Network with Conditional Cost Volume Normalization [33]. Noise-Aware Unsupervised Deep LiDAR-Stereo Fusion [34].

**Table 2 sensors-22-09364-t002:** Comparison of the methods that have been tested on the KITTI depth completion dataset showing the mean absolute error (MAE) and root mean squared error (RMSE), both in millimetres, and also the inverse mean absolute error (iMAE) and inverse root mean squared error (iRMSE), both in one/kilometres. The results with the highest accuracy are shown in bold.

	MAE (mm)	RMSE (mm)	iMAE (1/km)	iRMSE (1/km)
CCVN [33]	252.5	749.3	**0.8069**	**1.3968**
PENet [29]	209.0	757.2	0.9	2.22
SemAttNet [30]	205.49	709.41	0.90	2.03
VPN [32]	205.1	**636.2**	0.9870	1.8721
DySPN [31]	**192.71**	709.12	0.82	1.88

**Table 3 sensors-22-09364-t003:** Comparison of the methods that have been tested on the KITTI stereo dataset showing the percentage of pixels with an error greater than 3 pixels. The result with the highest accuracy is shown in bold.

	3.0 px
LiDARStereoNet ^1^ [34]	1.98
RAFT [26]	1.96
CREStereo [27]	1.69
LEAStereo [29]	**1.65**

^1^ LiDARStereoNet is evaluated on a subset of the stereo dataset containing 141 scenes out of 200.

**Table 4 sensors-22-09364-t004:** The strengths and limitations of each method.

Methods	Strengths	Limitations
Raft [26]	lower memory usage from indexing features model is available to download	only RGB sensors used lower accuracy
CREStereo [27]	lower memory usage from indexing features fast inference time of 0.41 s higher accuracy model is available to download	only RGB sensors used
LEAStereo [28]	fast inference time of 0.30 s higher accuracy model is available to download	higher complexity from using neural architecture search higher memory usage due to using 3D convolutions only RGB sensors used
PENet [29]	multiple sensor modalities used lower memory usage due to only using 2D convolutions very fast inference time 0.032 s model is available to download	lower accuracy
SemAttNet [30]	lower memory usage due to only using 2D convolutions multiple sensor modalities used very fast inference time of 0.2 s model is available to download	higher complexity from using attention-based modules lower accuracy
DySPN [31]	multiple sensor modalities used lower memory usage due to only using 2D convolutions very fast inference time of 0.16 s higher accuracy	model is not available to download
VPN [32]	multiple sensor modalities used higher accuracy	higher memory usage due to using 3D convolutions slow inference time of 1.4 s model is not available to download
CCVN [33]	multiple sensor modalities used higher accuracy	higher memory usage due to using 3D convolutions slow inference time of 1.011 s model is not available to download
LiDARStereoNet [34]	multiple sensor modalities used model is available to download	higher complexity from using an unsupervised training loss higher memory usage due to using 3D convolutions lower accuracy

## Data Availability

Data available in a publicly accessible repository.

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
