# Peer review of "A Critical Review of Deep Learning-Based Multi-Sensor Fusion Techniques"

_sensors, 2022, doi:10.3390/s22239364_

Round 1

Reviewer 1 Report

This article analyzes state-of-the-art fusion techniques which in recent years,and provides a detailed coverage of multi-sensor fusion techniques that use RGB stereo images and a sparse LiDAR projected depth map as input data to output a dense depth map prediction. I think this article has important reference value for researchers in this field.

Author Response

We thank this reviewer for their review and we have followed the feedback from the other reviewers in producing a revised version of the paper which can be seen as an attachment.

Reviewer 2 Report

In this paper, the authors provided “a review of multi-sensor fusion techniques” and “a new method for RGB-LiDAR fusion using confidence prediction”. However, as a review paper, it is not necessary to provide a new method. For example,

W.G. Hatcher and W. Yu "A Survey of Deep Learning: Platforms, Applications and Emerging Research Trends", IEEE Access, Vol. 6, pp. 24411 - 24432, 2018

https://ieeexplore.ieee.org/stamp/stamp.jsp?arnumber=8351898

Being a review paper, I suggest the authors to remove the “new method” parts for further review. Otherwise, please choose the “research article” for the reviewers to evaluate the proposed for the “new method”.

Author Response

We thank this reviewer for their feedback and have followed their suggestion of removing the new method parts from the paper and leaving it as a review paper. We have attached the new version of the paper where the changes are shown in blue.

Reviewer 3 Report

The paper presents a review of the existing work in multi-sensor fusion techniques and proposes a new fusion network for RGB-LiDAR.

Suggestions for improvement:

1. Since the title is "critical review", adding the strengths and limitations of each fusion technique / category is necessary.

2. Adding a skeleton of the techniques will be helpful in guiding the readers to follow the manuscript.

3. As the manuscript is more about the review, and the new proposed method has not outperformed the existing methods in most of the evaluation metric, suggesting to make this manuscript a pure review paper. Add more critical review / findings about the existing works. Add some summary table to give clearer idea to the readers.

The manuscript is well-written in overall. Thank you.

Author Response

We thank this reviewer for their feedback and have followed their suggestions for improvement. We have attached the new version of the paper where the changes are shown in blue.

Round 2

Reviewer 2 Report

This version of paper is modified as a well-written review paper. I agree the comments from other reviewers and suggest this paper to be accepted.